# Supercurrent mediated by helical edge modes in bilayer graphene

Prasanna Rout[1], Nikos Papadopoulos[1], Fernando Peñaranda [2],
Kenji Watanabe [3], Takashi Taniguchi [4], Elsa Prada [2], Pablo San-Jose [2] ✉ &
Srijit Goswami [1] ✉

Bilayer graphene encapsulated in tungsten diselenide can host a weak topological phase with pairs of helical edge states. The electrical tunability of this phase makes it an ideal platform to investigate unique topological effects at zero magnetic field, such as topological superconductivity. Here we couple the helical edges of such a heterostructure to a superconductor. The inversion of the bulk gap accompanied by helical states near zero displacement field leads to the suppression of the critical current in a Josephson geometry. Using superconducting quantum interferometry we observe an even-odd effect in the Fraunhofer interference pattern within the inverted gap phase. We show theoretically that this effect is a direct consequence of the emergence of helical modes that connect the two edges of the sample. The absence of such an effect at high displacement field, as well as in bare bilayer graphene junctions, supports this interpretation and demonstrates the topological nature of the inverted gap.

Helical edge modes in two-dimensional (2D) systems are an important building block for many quantum technologies, such as dissipationless quantum spin transport[1–3], topological spintronics[4,5], and topological quantum computation[6]. Electrons traveling along these modes cannot invert their propagation direction unless their spin is flipped. As a consequence, they cannot backscatter as long as time-reversal symmetry is preserved (e.g., even in the presence of arbitrary non-magnetic defects). Helical states are expected to appear at the edges of 2D topological insulators[7,8] or in one-dimensional semiconductors with large spin-orbit coupling (SOC)[6]. Interestingly, single-layer graphene was the first theoretically predicted quantum spin Hall insulator[9], whereby the intrinsic SOC gives rise to helical edge states. However, the strength of this Kane-Mele type SOC ($\lambda_{KM}$) is too small in graphene (≈40 μeV) to realize a topological phase in practice.

With the advent of graphene-based van der Waals heterostructures with exceptional electronic properties, several alternative approaches have been explored to create helical edge modes. These modes are shown to exist in the quantum Hall regime of single-layer graphene at filling factor $v = 0$ under the application of a large in-plane magnetic field[10], or when placed on a substrate with an exceptionally large dielectric constant[11]. Using a double-layer graphene heterostructure, helical transport was also observed by tuning each of the layers to $v = \pm 1$[12]. While these experiments did not involve superconductivity, they have been complemented by theoretical proposals showing that coupling the helical modes to a superconductor should give rise to topological superconductivity[13,14]. Unlike the experiments involving topological insulators[15–18], the main practical drawback of these proposals is the requirement of large magnetic fields, which is detrimental to any system involving superconductors.

Recently, it was shown that helical modes can appear at zero magnetic field in bilayer graphene (BLG) encapsulated with WSe$_2$, a transition metal dichalcogenide (TMD)[19]. Several experiments have shown that graphene coupled to TMDs gives rise to a proximity induced Ising-type SOC, denoted $\lambda_I$[20–27]. In the case of BLG symmetrically encapsulated in WSe$_2$ (Fig. 1a), $\lambda_I$ has opposite signs on the two graphene layers, thereby effectively emulating a Kane-Mele type SOC,

[1]QuTech and Kavli Institute of Nanoscience, Delft University of Technology, 2600 GA Delft, The Netherlands. [2]Instituto de Ciencia de Materiales de Madrid (ICMM), CSIC. Sor Juana Inés de la Cruz 3, 28049 Madrid, Spain. [3]Research Center for Functional Materials, National Institute for Materials Science, Tsukuba 305-0044, Japan. [4]International Center for Materials Nanoarchitectonics, National Institute for Materials Science, Tsukuba 305-0044, Japan. ✉e-mail: pablo.sanjose@csic.es; s.goswami@tudelft.nl

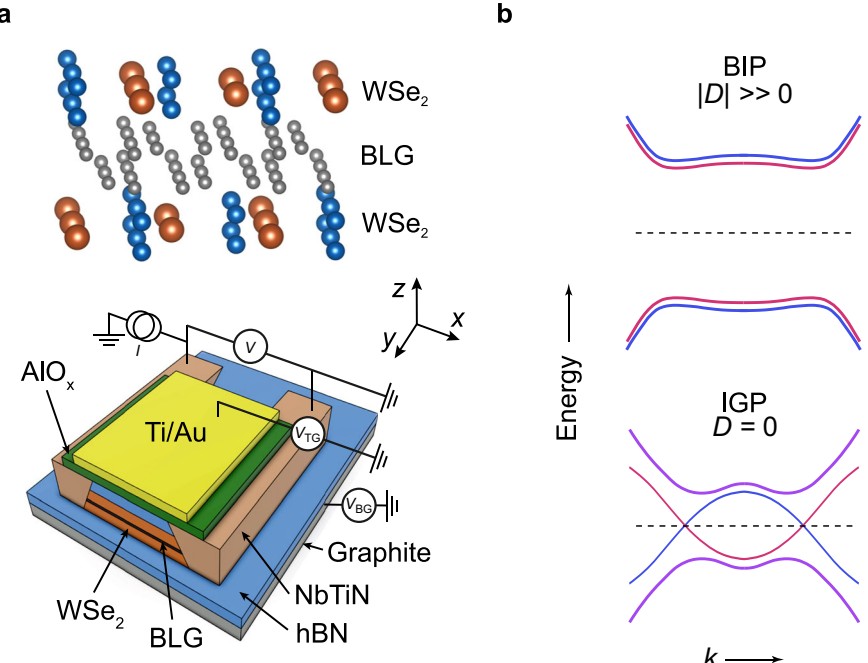

**Fig. 1 | Band inversion in a bilayer graphene junction. a** Device schematic of a NbTiN-based Josephson junction. The BLG is symmetrically encapsulated in WSe$_2$ leading to a proximity induced SOC. The hexagonal boron nitride (hBN) and hBN/AlO$_x$ act as bottom and top-gate dielectrics, respectively. The measurements are performed using a quasi-four terminal current-biased circuit, where $I$ is the current bias, $V$ is the measure voltage, and $V_{TG(BG)}$ is the voltage applied to top (bottom)

gate. **b** Band structure of the encapsulated BLG for different displacement fields $D$. At high $|D|$ ($|u| > \lambda_I$) the bulk is in a band-insulator phase (BIP), whereas the SOC creates an Ising gap in the bulk bands around $D = 0$, driving the system into an inverted-gap phase (IGP). In this weak topological phase, gapless edge states are present all around the BLG stack.

but with a significantly larger magnitude of few meVs[19,25,28]. Using an electric displacement field $D$ across the BLG, it was shown[19] that the band structure could be tuned continuously from a band-insulator phase (BIP) at large $D$, through a topological phase transition, and into an inverted-gap phase (IGP). In this work we combine such a WSe$_2$/BLG/WSe$_2$ heterostructure with a superconductor and study the Josephson effect across this phase transition. In the normal phase we replicate the previously measured[19] transition between a BIP and an IGP. In the superconducting phase we show that the IGP is characterized by a local minimum of the critical current $I_c$. In a topological IGP, theory predicts that currents should be able to circulate around the BLG sample via topological helical edge modes. We show theoretically that this should in turn cause a unique even-odd modulation with flux in the Fraunhofer interference pattern of the BLG Josephson junction (JJ). We present quantum interferometry measurements that exhibit this modulation. Importantly, the modulation is found to disappear upon entering the BIP of our encapsulated samples, and is altogether absent in trivial BLG control devices without WSe$_2$ encapsulation. Collectively, these observations suggest a topological nature for the superconducting IGP, and the presence of supercurrent-carrying topological helical modes.

## Results

A schematic of the devices is shown in Fig. 1a ("Methods" section and Supplementary Section 1 for details). The WSe$_2$/BLG/WSe$_2$ heterostructure has superconducting NbTiN contacts as well as a back gate, $V_{BG}$, and a top gate, $V_{TG}$. We present results on two devices (Dev A and Dev B) fabricated on the same heterostructure. Dev A has a larger contact separation, $L = 3.7\,\mu m$, which suppresses the Josephson effect and allows us to study the normal-state properties of the heterostructure. The supercurrent transport is explored in a ballistic JJ (Dev B) with a contact separation of $L \approx 300$ nm. Analogous measurements

on a third device Dev D are found to be consistent with the results for Dev B, see Supplementary Sections 5, 6 and 7.

Figure 1b shows how the BLG band structure is predicted to evolve as a function of displacement field $D$ in the presence of WSe$_2$-induced SOC. For $D = 0$ the valence and conduction bands of the BLG are inverted due to the Ising SOC. The band inversion opens up a gap $\lambda_I$ in the bulk band structure. Increasing $|D|$ introduces an additional competing energy $u = -edD/\epsilon_{BLG}$. Here, $d = 0.33$ nm is the BLG interlayer separation and $\epsilon_{BLG} = 4.3$ is the out-of-plane dielectric constant of BLG. At large $|D|$, $u$ becomes the dominant energy scale compared to $\lambda_I$ and a gap associated with layer-polarized bands opens up. Thus, by tuning $|D|$ one can transition from the inverted phase (IGP: $|u| < \lambda_I$) to band-insulator phase (BIP: $|u| > \lambda_I$) via an inversion point, where the bulk gap closes. The IGP is a weak topological insulator, with two associated helical modes per edge. If we neglect for the moment the possibility of additional trivial BLG/vacuum edge modes, the normal resistance for sufficiently large, pristine samples is then expected to be $R \approx h/4e^2 = 6.4\,k\Omega$ in the IGP, owing to the emergence of helical edge transport, and the exponential suppression of currents across the gapped bulk.

We now compare this theory expectation to experiment. Before proceeding to the superconducting regime of our devices, we characterize the various phases in the normal regime. As a first step we measure the dependence of normal resistance with bottom and top gates, which independently control the carrier density, $n$, and the displacement field, $D$ (Fig. 2a). The resistance $R$ as a function of $n$ and $D$ reveals a local maximum in $R$ close to $D = 0$ along the $n = 0$ line. This is consistent with the predicted opening of a SOC gap in the IGP ($\lambda_I > |u|$). A similar behavior has been observed previously in the form of an incompressible phase in capacitance measurements[19]. As we increase $|D|$, the gap is reduced and $R$ decreases, reaching a minimum at the gap-inversion point $\lambda_I = |u|$. Increasing $|D|$ further leads to an increase in

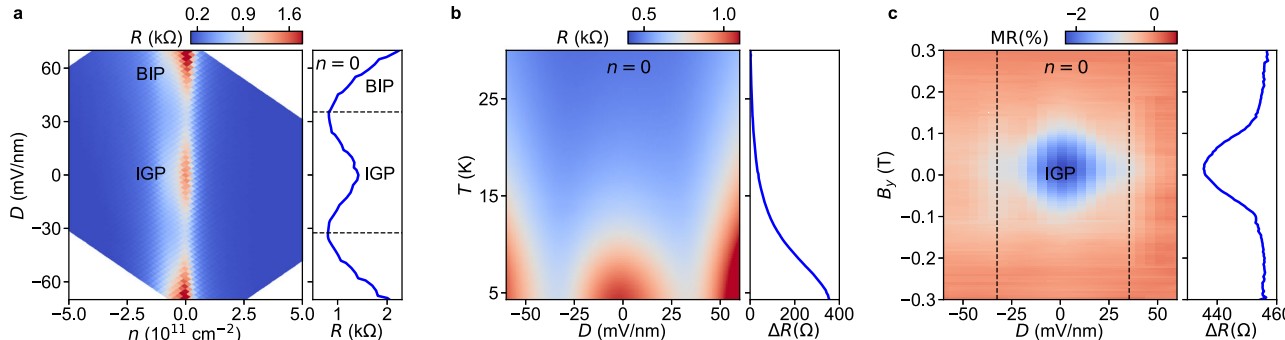

**Fig. 2 | Bulk and edge transport in the inverted-gap phase. a** Resistance $R$ at 3.3 K for Dev A measured as a function of carrier density $n = (C_{TG}V_{TG} + C_{BG}V_{BG})/e$ and displacement field $D = (C_{TG}V_{TG} - C_{BG}V_{BG})/2\epsilon_0$, where $C_{TG(BG)}$ is the capacitance of top (bottom) gate and $V_{TG(BG)}$ is the voltage applied to the top (bottom) gate. Side panel: line cut of the $R(n, D)$ map at $n = 0$. **b** Temperature dependence of resistance $R(T)$ measured at $n = 0$ for Dev A. Side panel: the disappearance of the IGP (between the two dotted lines) at higher temperatures is seen from the difference $\Delta R$ between $R$ at $D = 0$ and at the inversion point, $D_{inv} = -32$ mV/mm. **c** The magnetoresistance MR $= [R(B_y) - R(0.3\,T)]/R(0.3\,T)$ measured at 40 mK as a function of $D$ and in-plane magnetic field $B_y$ for Dev A. The dotted lines represent the inversion points. The observed dip close to $D = 0$ and $B_y = 0$ is the result of the conduction due to helical edge channels. Side panel: field dependence of $\Delta R$.

$R$ as the band-insulator gap grows. The inversion points corresponding to $R$ minima are at $D = -32$ and 35 mV/nm (Fig. 2a), which yield $\lambda_I = 2.5$ and 2.7 meV. The estimated $\lambda_I$ matches quite well with previously reported values of $2.2 - 2.6$ meV[19,25,26]. While this trend is qualitatively consistent with theory, one should note that the magnitude of the IGP resistance is smaller than expected. As we will argue below, we interpret this as the result of a finite bulk conduction, enabled by the charge-puddles in the BLG at small $D$, which is known to arise as a result of intrinsic disorder or defects[29-32] and of charge inhomogeneity from chemical or electrostatic doping[33-35]. The SOC gap can be probed by thermal excitation, see Fig. 2b. With increasing temperature, $R$ decreases as would be expected for a thermally activated gap, except that at low temperatures, $R$ is capped to the finite value from bulk conduction. In addition, the local $R(D)$ maximum at $D = 0$ gradually becomes less pronounced as the temperature is increased, and completely vanishes at higher temperatures. To see this effect more clearly we determine the difference in the resistances at $D = 0$ and at the inversion point $D_{inv} = -32$ mV/mm, i.e., $\Delta R = R(D = 0) - R(D = D_{inv})$, which acts as an indicator of band inversion. While $\Delta R$ is positive close to $D = 0$ in the IGP, it is almost zero at $T \approx 26$ K (side panel of Fig. 2b). The gap extracted from fitting the $R(T)$ dependence reveals the presence of a gap maximum around $D = 0$ and a gradual increase of the gap with increasing $D$ in the BIP (Supplementary Section 2).

While the temperature dependence provides information about the bulk gap, a verification of the existence and the helical character of edge states can be done by breaking time-reversal symmetry. An in-plane magnetic field $B_y$ introduces a Zeeman energy $E_Z$ which opens a gap in the helical edge states[19], and removes their contribution from conduction. To check this effect we measure $R$ at $n = 0$ in the presence of an in-plane magnetic field ($B_y$) as presented in Fig. 2c. At $D = 0$ we observed a dip in magnetoresistance MR $= [R(B_y) - R(0.3\,T)]/R(0.3\,T)$ for $B_y < 0.15$ T indicating the extra conduction from edge states. At higher fields the absence of these gapless states results in MR values close to zero. Outside the IGP where no helical edge is present, MR$\approx 0$ for all $B_y$. Additionally, we can rule out that this effect is related to the bulk bands as $E_Z < \lambda_I$. Moreover, the positive $\Delta R$ for the complete field range is consistent with a band inversion originating from the SOC gap (side panel of Fig. 2c). At lower fields we once again see the presence of conducting edge channels resulting in a dip in $\Delta R$. This behavior is consistent with the helicity of at least some of the edge modes in the IGP, as trivial spinful modes are not expected to be suppressed by $B_y$.

After checking the plausible helicity of IGP edge states, we turn our focus towards inducing superconductivity in these edge modes using JJs. The JJ (Dev B) is in the ballistic limit, as indicated by Fabry-Perot oscillations in the normal-state transport[36,37] (Supplementary

Section 3). We first check the evolution of the critical current ($I_c$) in the JJ as a function of $n$ for fixed $D = 0$ (Fig. 3a) and vice versa (Fig. 3b). The dependence of $I_c$ on $n$ is similar to several previous studies[36-40]. Figure 3a shows an $I_c$ minimum at $n = 0$, coinciding with the $R$ maximum, while $I_c$ increases with higher electron/hole doping. The more interesting behavior is the $D$ dependence of $I_c$ at $n = 0$, displayed in Fig. 3b. We observe a local minimum in $I_c$ at $D = 0$, corresponding to the local $R$ maximum, and two maxima at higher $|D|$ values, concurrent to the $R$ minima at the band inversion points. At higher $|D|$, $I_c$ decreases monotonically due to the gradual opening of a band-insulator gap[40].

The suppression of the supercurrent around $D = 0$ is a consequence of the higher resistance in the IGP. This suppression, however, does not reveal whether the supercurrent flows through the bulk or through edge channels. One could attribute the suppression of $I_c$ in the IGP to the opening of a bulk SOC gap entirely. Therefore, demonstrating the existence of proximitized edge modes requires a different probe that can differentiate edge from bulk supercurrents. A common method employed for this purpose is superconducting quantum interferometry (SQI)[15,16,18,39-42], which involves the measurement of $I_c$ as a function of a perpendicular magnetic field $B_z$. In the case of a short-and-wide JJ with homogeneous current transport across its width $W$, $I_c(B_z)$ should display a standard Fraunhofer pattern, following the functional form $I_c \sim \sin(\pi\Phi/\Phi_0)/(\pi\Phi/\Phi_0)$, where $\Phi = B_z/LW$ and $\Phi_0 = h/2e$ is the superconducting flux quantum. This regime is observed in our samples at high densities $n$ (see Supplementary Section 3, Figure S7). On the other hand, for a JJ with only two superconducting and equivalent edges (i.e., a symmetric SQUID) one expects a SQI pattern of the form $\cos(\pi\Phi/\Phi_0)$. However, the SQI pattern becomes more complicated if there is some coupling between these edges. An efficient inter-edge transport along edge channels flowing along the two SN interfaces (without becoming fully gapped by proximity) can lead to the electrons and holes flowing around the planar JJ, thereby picking up a phase from the magnetic flux. This phase introduces a $2\Phi_0$-periodic component into the $\Phi_0$-periodic SQUID pattern of the edge supercurrent[43], which then becomes a clear signature of the existence of proximitized NS edge modes connecting the vacuum edges.

Given its importance for the interpretation of our experimental SQI results presented below, we first provide a theoretical analysis of how $2\Phi_0$ harmonics arise in $I_c(\Phi)$ as a result of electrons and holes circulating around the sample. We compute $I_c$ in a simple model containing the minimal ingredients to capture the effect (see Supplementary Section 9 for further details). The JJ with a gapped central region is abstracted into just four sites; one per corner of the BLG region. The parent superconductor induces on the $j$-th corner a pairing

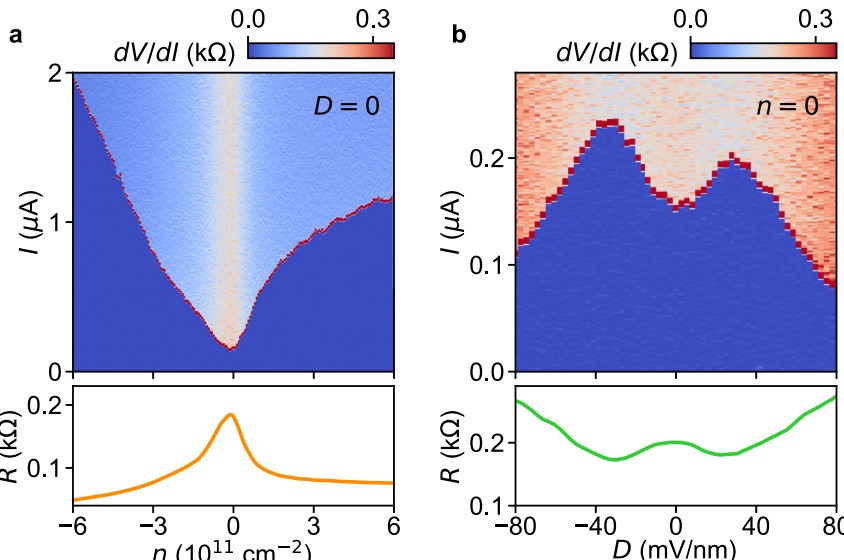

**Fig. 3 | Supercurrent in the inverted-gap phase. a** Current bias $I$ dependence of the differential resistance $dV/dI$ for Dev B as a function of carrier density $n$ at $D = 0$ and $T = 40$ mK. The jump in $dV/dI$ marks the critical current $I_c$. The lower panel shows the normal-state resistance $R(n)$ for $D = 0$. **b** $dV/dI$ (upper panel) and normal-state resistance $R$ (lower panel) as a function of $D$ for $n = 0$.

potential $\Delta e^{i\phi_j}$, see Fig. 4a. The pairing phases $\phi_j$ depend on the magnetic flux $\Phi$ through the BLG region, which is described by a gauge field $A_y = B_z x$. This makes the tight-binding Hamiltonian $H$ of the four sites strictly $\Phi_0$-periodic in $\Phi$. Next, we add two different hopping amplitudes between the corners, representing the existence of edge channels flowing around the central region. These hoppings acquire a Peierls phase induced by $A_y$. The horizontal hoppings along the vacuum edges are denoted by $t$. They have zero Peierls phase, and thus preserve the $\Phi_0$ periodicity of the Hamiltonian. The vertical hopping along the left NS interface is denoted by $\tau$, again with zero Peierls phase since it is located at $x = 0$. On the right NS interface at $x = L$ the hopping has the same modulus $\tau$, but its Peierls phase is now $e^{\pm i\pi\Phi/\Phi_0}$, depending on the direction. This term makes the Hamiltonian $2\Phi_0$-periodic when $\tau \neq 0$.

The critical current $I_c$ may be computed by maximizing the Josephson current $I = (2e/\hbar)\partial_\phi F$ versus $\phi$, where $F$ is the free energy computed from $H$. The resulting $I_c$, shown in Fig. 4b, therefore inherits the periodicity of $H$ with $\Phi$. For a small $\Delta/t$ and a small but finite $\tau/t$, the resulting critical current is accurately approximated by[43] $I_c \sim |\cos(\pi\Phi/\Phi_0) + f|$ for a positive $0 \leq f < 1$ that grows with $\tau$, see the Supplementary Sections 10 and 13. A finite $f$ results in an even-odd modulation of the $\tau = 0$ SQUID-like pattern. As $\tau$ approaches $t$, $I_c$ develops higher harmonics that deviate from this simple expression. The transport processes enabled by the coupling $\tau$ along the NS interfaces, i.e., by Josephson currents looping around the normal region, lead to the even-odd effect in the Fraunhofer pattern.

We may improve the minimal model by adopting a more microscopic description with several trivial transport channels flowing along vacuum edges[39,40], and two helical modes flowing all around the BLG junction, as expected from the band-inverted phase[19,28,44]. A coupling $\tau$ between vacuum and helical states enables an inter-edge scattering mechanism, see Fig. 4c. This time the Peierls phase is incorporated into the tight-binding discretization of the different modes. Like in the minimal model, the inter-mode coupling and the superconducting proximity effect are both assumed to take place at the corners of the normal sample, whose bulk is again assumed to be completely gapped. Although these simplifying assumptions are not strictly satisfied in the experimental samples, they are enough to confirm that the conclusions drawn from the minimal model still hold in a more generic situation, with propagating helical modes in place of a direct inter-edge hopping

and with multiple trivial vacuum edge modes. The results for $I_c$ in the case of one and two vacuum edge modes are shown in Fig. 4d, e. We once more find a $\Phi_0$-periodic SQUID-like pattern at $\tau = 0$ (in red), representative of the non-inverted phase. By switching on the coupling $\tau$, an even-odd modulation arises. This holds true also for higher number of vacuum modes. The main difference with respect to the minimal model is the behavior when $\tau \to t$, which is now less drastic.

Keeping these theoretical results in mind, we measure the SQI patterns as a function of $D$ at $n = 0$. Fig. 5a shows interference patterns at three different values of $D$. In contrast with the simulations, we observe a large central peak at $B_z = 0$. This is indicative of a finite bulk conduction, in line with our inference from the measurements of normal resistance. However, the SQI pattern at $D = 0$ (inside the IGP) shows a clear even-odd effect, i. e. the odd lobes are less intense compared to subsequent even lobes. In contrast, at high $D$ (inside the BIP) this effect is lost, giving rise to a more Fraunhofer-like pattern. Fig. 5c shows the evolution of the SQI as a function of $D$, where it becomes clear that the even-off effect is present only in the IGP and is strongest at $D = 0$. As $|D|$ is increased further (thus going deeper into the BIP), the intensity of $B_z = 0$ peak decreases, and a SQUID-like pattern without any even-odd effect emerges (Supplementary Section 7), indicating the presence of trivial and decoupled vacuum edge modes. To further confirm that the even-odd effect is associated with superconducting helical modes, we perform control experiments on a bare hBN/BLG/hBN JJ (Dev C), where we expect no induced SOC and hence no helical modes. Indeed, we do not observe any even-odd effect in the control SQI patterns although the trivial edges are still present at higher $D$ (Fig. 5e, f and Supplementary Section 7).

## Discussion

Even-odd modulated SQI patterns have been experimentally reported in various 2D systems with and without bulk conduction[16,18,41,42]. The modulation is qualitatively described by the addition of a flux-independent supercurrent offset in the standard interference pattern (Supplementary Section 13). In InAs- and InSb-based planar JJs[41,42], it was proposed that the even-odd effect (and the flux-independent offset) stems from crossed Andreev reflection connecting the trivial edges of the JJs via conducting NS interfaces. Unfortunately, its origin was not fully clarified in these experiments. In contrast, our symmetrically encapsulated BLG junctions offer a very natural candidate for an

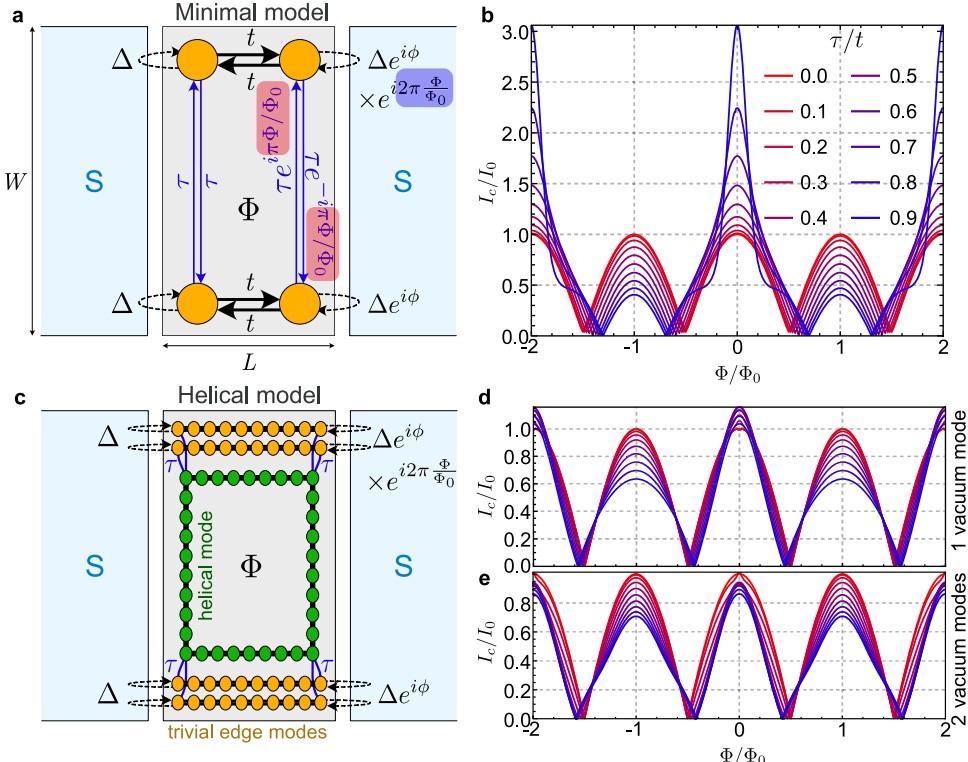

**Fig. 4 | Theory of the Fraunhofer even-odd effect from helical edge states.**
**a** Sketch of the four-site minimal model, in terms of the induced superconducting pairings at the corners (with amplitude $\Delta$, phase $\phi$, magnetic flux $\Phi$ and flux quantum $\Phi_0$), intra-edge hoppings $t$ and inter-edge hoppings $\tau$, with their Peierls phases in the gauge $A_y = Bx$. The origin of coordinates is chosen at the bottom-left corner. **b** Critical current $I_c(\Phi)$, normalized to a fixed $I_0$, of the minimal model exhibiting an even-odd modulation for $\tau \neq 0$. **c** Sketch of a more elaborate model, including in each spin sector one or more trivial edge modes (yellow) and one helical mode (green) flowing around a gapped bulk. **d, e** Corresponding $I_c(\Phi)$ for one and two trivial edge modes, respectively, exhibiting the even-odd effect when trivial and helical modes become coupled by a hopping $\tau$ at the corners.

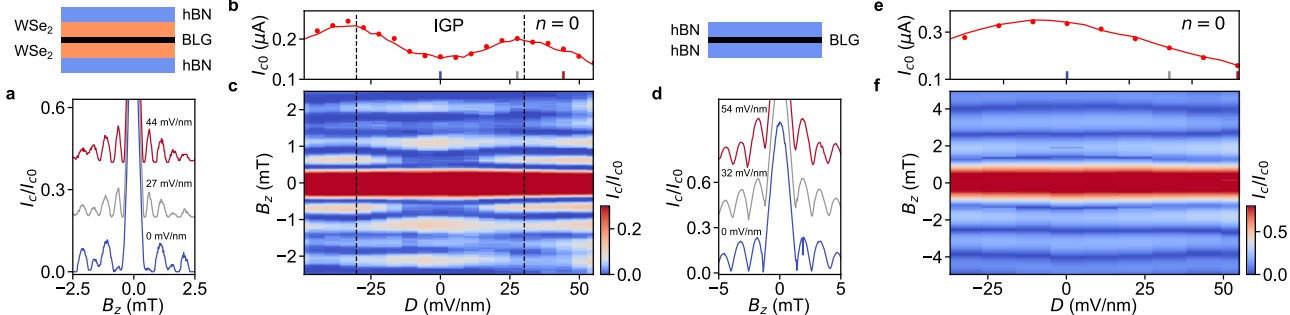

**Fig. 5 | Even-odd effect in superconducting quantum interferometry.**
**a** Normalized critical current $I_c/I_{c0}$ as a function of perpendicular magnetic field $B_z$ measured at three different displacement field $D$ for Dev B while keeping $n = 0$. The SQI patterns are vertically shifted by 0.2 from each other. **b** Zero-field critical current $I_{c0}$ extracted from Fig. 3b (solid line) and from (**c**) (circles) as a function of $D$. **c** Normalized critical current $I_c/I_{c0}$ as a function $B_z$ and $D$. An even-odd effect in the SQI oscillatory pattern can be observed only within the IGP (marked by two vertical dotted lines). **d**–**f** Same as (**a**–**c**) but for Dev C where a BLG is encapsulated in hBN without WSe$_2$. The solid line in (**d**) is the extracted supercurrent in the Supplementary Section 4.

efficient inter-edge coupling mechanism, in the form of helical modes flowing along the two NS boundaries in the IGP. The even-odd effect then acquires a special significance in our experiment, as a probe of the emergence of edge modes in the IGP. Their disappearance after the band inversion and their response to in-plane Zeeman fields offer complementary evidence of their helicity and the probable topological nature of the inverted gap, and serves as a demonstration of how their emergence can be controlled with an electric field.

Our work provides the first experimental evidence of supercurrent flow along helical edges in graphene. The clear $2\Phi_0$-periodic signature in our SQI experiment suggests an enticing prospect of detecting topological $4\pi$-periodic current-phase relations in this system[8], which requires the investigation of the a.c. Josephson effect[17,18,45] and/or non-equilibrium effects at finite bias in a.c. supercurrent[46]. The observation of gate tunable helical edge-mediated supercurrents opens a promising new avenue[44] towards topological superconductivity and Majorana physics in van der Waals materials.

## Methods

### Fabrication
We exfoliate flakes of bilayer graphene (BLG), graphite (5–15 nm), tungsten diselenide WSe$_2$ (10–35 nm) and hexagonal boron nitride

hBN (20–55 nm) on different $SiO_2/Si$ substrates using Scotch tape. Then the stacks of hBN/WSe₂/BLG/WSe₂/hBN/graphite are assembled using the van der Waals dry-transfer using polycarbonate (PC) films on Polydimethylsiloxane hemispheres. The flakes are picked up at 110 °C. The stacks are checked with atomic force microscopy before further fabrication.

To fabricate the Josephson junctions (JJs) we spin coat a bilayer of 495 A4 and 950 A3 PMMA resists at 4000 rpm and bake at 175 °C for 5 min, after each spinning. After the e-beam lithography patterning, the resist development is carried out in a cold $H_2O$:IPA (3:1) mixture. We perform a reactive ion etching step using $CHF_3/O_2$ mixture (40/4 sccm) at 80 μbar with a power of 60 W to etch precisely through the top hBN and WSe₂. Then we deposit NbTi (5 nm)/ NbTiN (110 nm) by dc sputtering for superconducting edge contacts and lift-off in NMP at 80 °C. In order to isolate the top-gate metal from the ohmic, we deposit a 30 nm dielectric film of $AlO_x$ using atomic layer deposition at 105 °C. Finally we define the top gate by e-beam lithography, deposit Ti (5 nm)/Au (120 nm), followed by lift-off.

### Measurements

All measurements are performed in a dilution refrigerator with a base temperature of 40 mK. Four-probe transport measurements are performed with a combination of DC and AC current-bias scheme. To determine critical current we have employed a voltage switching detection method with current source and critical current measurement modules[47]. The magnetic fields are applied by a 3D vector magnet, which enables us to align the field within ± 5° accuracy.

### Four-site model

The spinless BdG Hamiltonian used for the four-site model, written in the Nambu basis $(\boldsymbol{c}_\sigma, \boldsymbol{c}_{\bar{\sigma}}^\dagger)^T$ where $\boldsymbol{c}_\sigma = (c_{1\sigma}, c_{2\sigma}, c_{3\sigma}, c_{4\sigma})$, reads

$$H = \begin{pmatrix} H_0 & H_\Delta^+ \\ H_\Delta & -H_0^* \end{pmatrix}, \quad (1)$$

where

$$H_0 = \begin{pmatrix} 0 & t & 0 & \tau \\ t & 0 & \tau e^{-i\pi\Phi/\Phi_0} & 0 \\ 0 & \tau e^{i\pi\Phi/\Phi_0} & 0 & t \\ \tau & 0 & t & 0 \end{pmatrix}, H_\Delta = \Delta \begin{pmatrix} 1 & 0 & 0 & 0 \\ 0 & e^{i\phi} & 0 & 0 \\ 0 & 0 & e^{i\phi+i2\pi\Phi/\Phi_0} & 0 \\ 0 & 0 & 0 & 1 \end{pmatrix}. \quad (2)$$

$H_0$ and $H_\Delta$ are $4 \times 4$ matrices corresponding to the normal Hamiltonian of the particle sector and the onsite superconducting electron-hole pairing terms induced by the leads, respectively. $\Delta$ and $\phi$ are the induced pairing amplitude and phase from the parent superconductors, respectively. $\Phi$ is the flux into the normal region encompassed by the four sites. $t$ and $\tau$ are the hopping amplitudes between sites along the vacuum edge interfaces (1↔2 and 3↔4) and the NS interfaces (1↔4 and 2↔3), respectively. Further details, including the multimode generalization of this model, are given in Supplementary Sections 9 and 12.

### Data availability

Raw data and analysis scripts for all presented figures are available in the Zenodo database under accession code https://doi.org/10.5281/zenodo.10046824.

### Code availability

All computer codes employed in this work are available in the Zenodo database under accession code https://doi.org/10.5281/zenodo.7941266. Julia scripts can be executed after instantiating the package environment. Mathematica notebooks can be run directly from the top using version 13 and above.

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

## Acknowledgements
We thank Joshua Island for discussions regarding heterostructure assembly and Tom Dvir for comments on the manuscript. This research was supported by an NWO Flagera grant and TKI grant of the Dutch Topsectoren Program [P.R., N.P., S.G.], by the Spanish Ministry of Economy and Competitiveness through Grant PCI2018-093026 (FlagERA Topograph) [F.P., E.P., P.S-J.], by MCIN/AEI/10.13039/501100011033 and "ERDF A way of making Europe" through grant PID2021-122769NB-I00 [F.P., E.P., P.S-J.], and by the Comunidad de Madrid through Grant S2018/NMT-4511 (NMAT2D-CM) [E.P.].

## Author contributions
P.R and N.P. fabricated the devices, performed the measurements, and analyzed the experimental data. S.G. conceived and coordinated the experiment. T.T. and K.W. provided hBN crystals. F.P., E.P., and P.S-J. developed the theoretical interpretation and models, and F.P. performed the simulations. All authors participated in discussions and in writing of the manuscript.

## Competing interests
The authors declare no competing interests.
