## [Peer Review File · Nature Communications]

REVIEWER COMMENTS

Reviewer #1 (Remarks to the Author):

The manuscript reports interesting work on bilayer-graphene-WSe₂ samples. The motivation is to achieve topological superconductivity by inducing pairing correlations in helical edge modes that are claimed to occur in these samples. This is a very timely topic.

The manuscript is very well structured and contains very clear data presentation, but I do not find that the data adequately support the conclusions.

1. In Fig. 2, I may have missed this, but why is the resistance so low in the gapped phases? In the BIP phase it should be very large, while in the IGP phase it should be approximately one inverse quantum of conductance, so about 25 kOhm. Fig. 2c shows the MR to modulate resistance by only ~20 Ohm on a baseline of 400 Ohm in the gapped state at finite B_y . This suggests that transport is dominated by bulk or by many trivial edge modes.

Also, it would be nice to see absolute resistance rather than percentage MR in Fig. 2c.

2. On p. 3, the manuscript states that "This phase introduces

a $2\Phi_0$ -periodic component into the Φ_0 -periodic

SQUID pattern of the edge supercurrent I_s , which is a

highly significant signature of the existence of proximitized

helical modes." I am not sure I agree with this. Ref. 33 discussed that even-odd periodicity in the superconducting quantum interference pattern can be associated with trivial edge modes.

3. Fig. 5 is the key figure, where the even-odd effect associated with the claimed topological edges is supposed to be demonstrated. However, I find it very hard to ascertain if there is an even-odd effect based on the presented data.

Also, Figs. 5b,c,e all show a large supercurrent peak at $B=0$. This looks more like bulk transport than edge-dominated transport.

If the authors feel like there are simple explanations for these points, it would be very useful to see these, but as it stands, I am afraid I do not find the key claims to be supported by the data.

Reviewer #2 (Remarks to the Author):

The authors report a study of Josephson junctions on bilayer graphene encapsulated in WSe₂. The bilayer graphene in WSe₂ exhibits an inverted band gap at charge neutrality with helical edge transport. They succeeded in fabricating remarkable Josephson junctions equipped with top-gate electrodes, as well as a control experiment with Josephson junctions fabricated with bilayer graphene encapsulated in hBN.

The authors observed a clear change in the Fraunhofer pattern by tuning the WSe₂/BLG in the inverted bandgap regime. They observed the even-odd effect which is a partial suppression of the odd lobes. The data is of high quality and sample fabrication is very difficult. This inverted band gap phase of bilayer graphene is a very interesting and promising platform for inducing topological superconductivity. As such, I think the data really deserves to be published.

On the other hand, I have serious reservations about the authors' claims, which seem to me to be clearly exaggerated. Given the importance and history of the subject, I cannot recommend publication without significantly tempering the following claims about helical edge transport.

On page 2, second column, the authors claim that the small change in resistance near $D \sim 0$, together with the "activated" temperature dependence of the resistance there, provide evidence for the presence of helical edge states. I'm afraid this is not compelling evidence at all. Why does the resistance not reach $h/2e^2$? The authors claim that bulk transport is activated, but where is the Arrhenius plot that shows this? How can it be activated with such a small change in resistance? I notice in Fig. 2b that the ΔR vs T plot seems to saturate at low T , at a value that is still far from the expectation of helical edge transport. How do the authors interpret this? Residual bulk transport? How can we rule out that

this is not contributing to the supercurrent?

So I think the claim about helical edge transport, particularly the sentence "After checking for the presence of helical edge states, ..." and others, should be replaced by a more moderate presentation of the data, with a clear presentation of what can be concluded with certainty, and what cannot. This will not diminish the interest and importance of this work. On the contrary, a more factual and sincere analysis of this platform will be appreciated by the community and will allow us to move forward.

With regard to figure 3 and the corresponding description paragraph. I don't understand the sentence "A higher D , I_c decreases monotonically due to a gradual opening of a bandgap". Firstly, there is no bandgap signature in the data. The resistance is 2 kOhms, and I see no evidence of activation (Arrhenius plot?). Secondly, aren't we simply observing a constant $I_c R_n$, irrespective of the physics involved in varying the resistance?

The sentence "The presence of a supercurrent and its suppression at $D=0$ is a signature of Josephson coupling across a BLG in an inverted phase". I'm sorry, that's not correct. This observation is just the signature of a constant $I_c R_n$. It in no way indicates "Josephson coupling through a phase-reversed BLG". The causality is wrong.

Overall the BLG system is definitely interesting and the Fraunhofer data are beautiful. I strongly encourage the authors to modify their writing with more humble claims regarding the signature/presence of helical edge states, which I think weakens the paper.

Few more comments:

- In the paragraph "Before proceeding to the superconducting [...]" I suggest to add the reference Fig.2a, which appears only at the end, earlier in the paragraph.
- Figure 5: I suggest to plot the amplitude of each lobes as a function of D for both cases.
- Concerning the effective area and the model, I understand even-odd effect implies a $2\phi_0 = h/e$ periodicity of the Fraunhofer pattern due to a cross Andreev conversion along the superconducting contact (Fig S8). The theoretical part explains that this circular trajectory picks up an Aharonov-Bohm phase (so the $h/2$ periodicity). Then I would expect that this component of quantum interference has the effective area equal to the BLG area and not the larger effective area due to the magnetic penetration depth. Can the authors comment on that?

Reviewer #3 (Remarks to the Author):

Rout et al. investigate a Josephson junction (JJs) of bilayer graphene encapsulated in WSe₂ with top and bottom gates. In the normal state bilayer graphene in contact with WSe₂ has previously been shown to host a helical edge state due to Ising spin-orbit. Here, the authors first investigate the normal state properties of a large device showing that, close to charge neutrality, a finite displacement field results in an initial decrease followed by an increase in resistance of the device, which is characteristic of the bulk band inversion. They also show that an in-plane field modifies the magnetoresistance only for displacement field $D \approx 0$. Moving to JJs, Rout et al. show that the critical current at charge neutrality exhibits a two peak structure as a function of displacement field, again characteristic of the bulk band inversion. Finally, the main result of the paper is that the Fraunhofer pattern of the JJs show an even-odd pattern only for $D \approx 0$, which the authors ascribe to the presence of a helical edge. This is backed up by numerical simulations and by a control experiment on a device that is not encapsulated in WSe₂ and which does not show this even-odd pattern.

Overall I find that the results are very well written, logically presented, and interesting. The topic of inducing superconductivity in helical or chiral edges is of significant recent interest and the paper invites many avenues for investigating the nature of the induced superconductivity in this setup. I am minded towards eventual publication in Nat. Comm., but I reserve final judgment until the authors have responded to the following points:

1) My main concern is that the key result, the odd-even pattern of the Fraunhofer in a ballistic JJ with IGP, is only shown for a single device (dev B). I appreciate that fabrication can be time consuming, but I would be a lot more comfortable if the results were reproduced on (at least) a second device. As it currently stands I do not think I can recommend acceptance of a paper based on results from only a single device, even if the current results are very convincing.

2) In Fig. 2c the authors show the magnetoresistance is negative only close to zero in-plane magnetic field and zero displacement field, arguing that this is due to the opening of a gap in the helical dispersion due to the field. However, the effect is very small (~2%).

a) Can the authors comment on the small size of this effect? Naively one would expect a large increase in resistance due to the opening of a gap unless there are many other conduction channels than just the helical edge.

b) I appreciate that Fig. 2 is a larger device, but if there are several other conduction channels, why is the size of the Fraunhofer even-odd effect in dev B so large (almost killing the second lobe at $D=0$)? Does this not contradict somewhat the theoretical analysis in Fig. 4d-e that shows only a modest even-odd effect when several channels are present.

c) It would be useful (and convincing) to include in the supplemental material a similar plot to Fig. 2, but at a very small but finite density, such that the doping of the helical edge is finite and the chemical potential larger than any induced magnetic gap. In such a scenario one would expect very little change in MR.^[1]_{SEP}

3) In Fig. 3 it is shown that the displacement field results in two peaks in the critical current, I_c , which correspond to the expected transition from IGP. What is the impact of an in-plane magnetic field on I_c ? In particular within the IGP.^[1]_{SEP}

4) On a similar note, since the application of an in-plane field gaps out the helical modes it should therefore remove the even-odd pattern in the Fraunhofer lobes. Is this the case? Such a restoration of the usual Fraunhofer pattern would be a very convincing demonstration of the nature of the even-odd pattern stemming from the helical edge.

^[1]_{SEP}5) In Fig. S7 comparing the Fraunhofer at large and zero displacement field it is clear that the BLG (Fig. S7f) exhibits a shift of the lobes as function of displacement field (which is also visible in Fig. S7e). However, this shift does not appear for the WSe₂ encapsulated sample. Can the authors comment on what might have caused the shift for the BLG sample, but why it is not present for the sample with an IGP?

Minor comment: In Fig. 4b and e the colour map is dominated by the central lobe of the Fraunhofer pattern, which makes it a bit difficult to observe the main feature of interest (the side lobes). I would recommend setting adjusting this map to emphasise the side lobes.

We wish to thank all three referees for their thoughtful reports. A point by point response to all their comments and criticisms are provided below, together with a description of the associated changes in the manuscript.

REVIEWER 1

The manuscript reports interesting work on bilayer-graphene-WSe2 samples. The motivation is to achieve topological superconductivity by inducing pairing correlations in helical edge modes that are claimed to occur in these samples. This is a very timely topic.

The manuscript is very well structured and contains very clear data presentation, but I do not find that the data adequately support the conclusions.

We thank the referee for the appreciative comments. To strengthen our conclusions and claims we have extended our results with additional experimental data, see below.

1. In Fig. 2, I may have missed this, but why is the resistance so low in the gapped phases? In the BIP phase it should be very large, while in the IGP phase it should be approximately one inverse quantum of conductance, so about 25 kOhm.

This comment touches on a critical issue in our samples. As the reviewer points out, an ideal situation with a fully gapped bulk, and a long and wide enough pristine sample, all the current, both in the normal and the superconducting phase, should flow strictly along the edges of the BLG region. In the BIP phase the vacuum edges could carry a number N of trivial modes, while in the IGP phase two extra helical modes are added to each edge. If $N = 0$, one should then indeed expect a very large normal resistance in the BIP (exponential in the length of the sample L), while in the IGP, the resistance should be that of the four additional helical modes (two per edge). Ideally, then, we should go from an insulating R in the BIP, to an R minimum around the band inversion point, finally increasing to $R \approx h/4e^2 = 6.4\text{k}\Omega$ in the IGP phase.

In our samples the values of R are considerably lower than this. The resistance for Dev A is 2.1 k Ω in the BIP ($D = 70$ mV/mm, see Fig. 2 in main text), which corresponds to a sheet resistance of 4 k Ω . The sheet resistance in the IGP is 2.7 k Ω (See Fig. 1). This

FIG. 1. The sheet resistance R_S measured at $D = 0 = n$ for Dev A, B and D. The dotted lines represent the theoretically predicted value of $h/4e^2$.

suggests that a substantial deviation from the ideal $N = 0$ edge-only transport is taking place. We see two possibilities: that N is not zero, but rather around $N = 3$ (assuming a high trivial mode transmission), or that a finite current is flowing through the bulk. From our data analysis, we believe both to be the case in our samples. The presence of charge puddle disorder in the BLG region, a common occurrence in this kind of heterostructures, could be enabling hopping between puddles across the bulk. This is consistent with the Fraunhofer pattern in the superconducting phase, which shows a large peak around $B_z = 0$, instead of a pure SQUID-like pattern, at least for $D < 150\text{mV/nm}$. At higher displacement fields (deep BIP), the puddles are gapped out, and a SQUID-like pattern then emerges (see supplementary Fig. S11), consistent with a number of trivial edge states along the vacuum edges. Additionally, the presence of an even-odd modulation in the IGP suggests that the puddle disorder does not percolate through the sample, so that helical interedge channels can still emerge after the band inversion.

Given the importance of this observation for the interpretation of our measurements, we have included a brief version of the above discussion after presenting the normal resistance results. The sheet resistance values at $D = 0 = n$ is presented in supplementary section 6.

Fig. 2c shows the MR to modulate resistance by only 20 Ohm on a baseline of 400 Ohm in the gapped state at finite B_y . This suggests that transport is dominated by bulk or

FIG. 2. The resistance R and magnetoresistance $MR = [R(B_y) - R(0.3 \text{ T})]/R(0.3 \text{ T})$ measured at 40 mK as a function of D and in-plane magnetic field B_y for Dev A. The dotted lines represent the inversion points.

by many trivial edge modes. Also, it would be nice to see absolute resistance rather than percentage MR in Fig. 2c.

Indeed, as explained above, we agree that the smallness of the MR modulation is caused by bulk conduction background. We note, however, that the MR dip around $B_y = 0$ is clearly visible despite this background, and is consistent with the Zeeman-induced gapping of the helical modes in the IGP phase, as the dip vanishes for D away from the IGP. We also note that the magnitude of the MR modulation could be limited by the disorder along the edges, since the Zeeman gaps in the helical modes are expected to open precisely around neutrality, so that potential disorder along the edges could hinder the ability of Zeeman fields to suppress helical mode transport.

Since the resistance map (unlike the MR) cannot be used to directly detect the disappearance of helical mode transport [this was also an issue in Nature 571, 85–89 (2019)], we have chosen to leave the MR plot in the main text, but we have now added the resistance map to the supplementary (section 2C). We include both here also for the convenience of the reviewer.

2. On p. 3, the manuscript states that "This phase introduces a $2\Phi_0$ -periodic component into the Φ_0 -periodic SQUID pattern of the edge supercurrent [35], which is a highly significant signature of the existence of proximitized helical modes." I am not sure I agree with this. Ref. 33 discussed that even-odd periodicity in the superconducting quantum interference pattern can be associated with trivial edge modes.

Since the even-odd modulation only requires the existence of vertical channels connecting the two vacuum edges, an even-odd signal, by itself, is indeed no proof of topology or helicity of the inter-edge modes. This is the case of the even-odd effect observed for InAs junctions [Phys. Rev. Lett. 120, 047702 (2018) or Ref. 33 in previous version of the manuscript], where no topological modes are expected, and trivial edge modes are shown to exist. In our experiment, however, there is a crucial difference: we can connect and disconnect the even-odd modulation at will using a displacement field (Fig. S11 b,c). Moreover, the emergence of the modulation coincides with onset of the IGP, as seen both in the zero-field critical current and in the normal resistance. The observations are now reproduced in a second device (new sections 5 and 7 of the supplementary information). Importantly, our non-encapsulated control device did not exhibit even-odd modulation (Fig. 5d-f in the manuscript). Also, breaking time-reversal symmetry with an in-plane Zeeman field disrupts edge mode transport within the IGP. While it could be argued that all this is *still* not a direct proof of edge mode helicity and topology, it is "highly significant" evidence, collectively, so we had chosen to use that wording. All our observations are explained most naturally in terms of topological edge modes in the IGP, and none contradict this. We are therefore fully convinced that we are providing the right interpretation. We note in particular that, unlike the references: Phys. Rev. Lett. 120, 047702 (2018) and Phys. Rev. Res. 1, 032031 (2019), we observe SQUID-like oscillation deep in the BIP (Fig. S11 b,c), implying the presence of trivial vacuum edge states. However, these modes do not produce an even-odd effect in that regime (they are tied to the vacuum edges, so they induce no inter-edge transport). Therefore our even-odd modulations in the IGP must have a different origin than in those two works.

We have reworked our presentation in the main text to convey the above reasoning more precisely. We now use a more toned-down "complementary evidence" phrase, in place of "highly significant evidence" wording.

FIG. 3. The SQUID function $\mathcal{I}(\Phi) + f$ for a symmetric SQUID with uncoupled (a) and coupled edges (c). (b, d) Lower panels show the corresponding critical currents for $I_{c,0} = 1$. The SQUID function for a homogeneous JJ with uncoupled (e) and coupled edges (g). The corresponding critical currents for $I_{c,0} = 1$ are shown in the panels (f) and (h).

3. Fig. 5 is the key figure, where the even-odd effect associated with the claimed topological edges is supposed to be demonstrated. However, I find it very hard to ascertain if there is an even-odd effect based on the presented data. Also, Figs. 5b,c,e all show a large supercurrent peak at $B=0$. This looks more like bulk transport than edge-dominated transport.

We have now adjusted the color scheme of Fig. 5 and optimized the waterfall plots to make the even-odd effect more clear. In Fig. 5a,c we can clearly see that this effect is present in SQUID patterns for D close to zero. The odd lobes for these patterns have lower intensity compared to next even lobes.

As discussed previously, the residual bulk conductance indeed leads to a large central supercurrent peak. However this additional effect doesn't completely mask the even-odd interference originating from the supercurrent carried by helical edge states. Let us briefly clarify how that works. At a simplified level, we can understand its impact on the standard Fraunhofer and SQUID patterns as follows. As proposed by de Vries *et al.* [Phys. Rev. Res. 1, 032031 (2019)], the even-odd effect can be qualitatively modelled by a flux-independent

supercurrent offset f (that encodes the coupling between the two vacuum edges) in the SQI, i. e. $I_c(\Phi) \propto |\mathcal{I}(\Phi) + f|$. For a homogeneous JJ with uniform current flow $\mathcal{I}(\Phi) = \sin(\pi\Phi/\Phi_0)/(\pi\Phi/\Phi_0)$, while in the SQUID regime $\mathcal{I}(\Phi) = \cos(\pi\Phi/\Phi_0)$ [see Eq. (5) in the supplementary material]. We show the resulting SQI patterns for $f = 0$ (without the edge states or uncoupled edges) and $f = 0.05$ (finite coupling between the edge states) in Fig. 3. We see that in the homogeneous JJ case, the even-odd modulation produced by the edge modes is non-periodic, but still clearly visible, as in our experiment (Fig. 5 of the main text). Similar SQI patterns with both large central peak and even-odd modulation have been reported for InSb [Phys. Rev. Res. 1, 032031 (2019)] and HgTe [Nat. Nanotechnol. 12, 137–143 (2017)] JJs previously.

We have now clarified in the text the significance of the $B_z = 0$ peak in relation to background bulk transport.

If the authors feel like there are simple explanations for these points, it would be very useful to see these, but as it stands, I am afraid I do not find the key claims to be supported by the data.

We trust the improved discussion, and in particular the clarification of the role of the bulk contribution in our MR and SQI measurements, may convince the reviewer that our interpretation of the experimental data is solid. We have also added measurements from an additional device that shows the same overall even-odd and band inversion phenomenology, further strengthening the case for the topological helical mode interpretation.

REVIEWER 2

The authors report a study of Josephson junctions on bilayer graphene encapsulated in WSe2. The bilayer graphene in WSe2 exhibits an inverted band gap at charge neutrality with helical edge transport. They succeeded in fabricating remarkable Josephson junctions equipped with top-gate electrodes, as well as a control experiment with Josephson junctions fabricated with bilayer graphene encapsulated in hBN. The authors observed a clear change in the Fraunhofer pattern by tuning the WSe2/BLG in the inverted bandgap regime. They observed the even-odd effect which is a partial suppression of the odd lobes. The data

is of high quality and sample fabrication is very difficult. This inverted band gap phase of bilayer graphene is a very interesting and promising platform for inducing topological superconductivity. As such, I think the data really deserves to be published.

On the other hand, I have serious reservations about the authors' claims, which seem to me to be clearly exaggerated. Given the importance and history of the subject, I cannot recommend publication without significantly tempering the following claims about helical edge transport.

We thank the reviewer for the positive comments, and acknowledge her/his criticism about the way we expressed our claims. We have carefully revised the manuscript to tone down these claims, and more clearly state the limitations of our analysis (details below).

On page 2, second column, the authors claim that the small change in resistance near $D \sim n \sim 0$, together with the "activated" temperature dependence of the resistance there, provide evidence for the presence of helical edge states. I'm afraid this is not compelling evidence at all. Why does the resistance not reach $h/2e^2$? The authors claim that bulk transport is activated, but where is the Arrhenius plot that shows this? How can it be activated with such a small change in resistance? I notice in Fig. 2b that the Delta R vs T plot seems to saturate at low T, at a value that is still far from the expectation of helical edge transport. How do the authors interpret this? Residual bulk transport?

Thank you for this criticism. We agree that our analysis was too simplified, and crucial aspects such as the contribution of bulk transport and how it affects our conclusions deserve a much more careful discussion.

The issue of resistance quantization is an important one. Indeed, if the BLG were sufficiently wide, long and clean, without any trivial edge modes in the BIP, one would expect (a) a truly insulating BIP, and (b) a quantized $R = h/4e^2 = 6.4\text{k}\Omega$ IGP resistance (since the helical modes come in pairs, so there are two per edge). In our samples the values of R are considerably lower in general. The resistance for Dev A is 2.1 k Ω in the BIP ($D = 70$ mV/mm, see Fig. 2 in the manuscript), which corresponds to a sheet resistance of 4 k Ω , which then goes down to 2.7 k Ω in the IGP.

Therefore, a significant deviation from the ideal edge-only transport scenario is taking place. Two possible interpretations immediately come to mind: (a) there are N additional

trivial channels (both in the IGP and BIP) localized around the vacuum edges due to band bending (a common occurrence in graphene flakes), and/or (b) there is finite transport through the BLG bulk, whose gap is not enough to suppress conduction, perhaps due to charge puddles disorder (another known feature of graphene systems). Taking all our measurements together we conclude that both (a) and (b) are true. The evidence for (b) is that SQI critical current measurements show a strong $B_z = 0$ peak, which is associated to a uniform component to the current density across the Josephson junction. This is true for displacement fields D smaller than 150 meV/nm. Above that, the bulk gap is strong enough to completely shut down bulk transport, and a SQUID-like SQI develops, which is evidence for (a).

At smaller D , furthermore, bulk transport can occur in two different fashions: (1) complete shunting (the puddle disorder is so extensive that they percolate, the bulk behaves as a conductor, and no significant difference between the IGP and BIP is expected at low D); or (2) hopping between non-percolating puddles (so that helical edge modes are still present, and the IGP remains topologically distinct from the BIP, harboring helical modes). Since we observe clear evidence of a IGP-BIP band inversion (in resistance and critical current), plus the concomitant emergence of an even-odd modulation in the SQI, it seems clear that of these two possibilities our devices are in case (2).

Since this type of analysis is crucial to justify our claims of topological helical-modes in the IGP, we have included a more careful discussion of the above arguments in the revised text. We have also included Arrhenius plots for the IGP, and adjusted the corresponding activated-transport discussion in the main text to account for bulk puddles.

How can we rule out that this is not contributing to the supercurrent? So I think the claim about helical edge transport, particularly the sentence "After checking for the presence of helical edge states, ..." and others, should be replaced by a more moderate presentation of the data, with a clear presentation of what can be concluded with certainty, and what cannot. This will not diminish the interest and importance of this work. On the contrary, a more factual and sincere analysis of this platform will be appreciated by the community and will allow us to move forward. With regard to figure 3 and the corresponding description paragraph.

We agree, our manuscript should more carefully present the reasoning that, in our opinion, demonstrate the existence of helical modes in our measurements. We have revisited the discussion in the updated manuscript, and significantly toned down sentences like the above (which now reads “After checking the plausible helicity of IGP edge states ...”).

Please allow us to expand here on the rationale behind our main claims. We indeed cannot distinguish the effect of bulk and edge states from the $B = 0$ critical current measurements presented in the manuscript (Fig. 3). To detect the presence of helical edge states of topological origin (flowing around the BLG sample in the IGP, and hence connecting the two vacuum edges), and to distinguish their contribution to the supercurrent from that of bulk transport, we need a full SQI measurement (Fig. 5). Normal magnetoresistance is an important check, but SQI is the tool of choice here, since it is sensitive to the spatial distribution of the current. The presence of bulk conduction is revealed by the large central $B_z = 0$ peak (as opposed to the SQUID oscillation expected for fully insulating bulk). We now explicitly discuss this. Additionally, the effect of the inter-edge transport is revealed by the superimposed even-odd modulation, which is a unique consequence of interfering electron-hole paths that flow around the sample.

The even-odd modulation, however, does not reveal by itself the topological nature of the transport channels connecting the two vacuum edges, only their spatial location (along the two vertical NS interfaces). The evidence of their topological nature is twofold: (1) the even-odd signal emerges precisely as the sample transitions into the IGP, as measured in normal resistance, magnetoresistance and critical current, and (2) the presence of trivial vacuum edge states (revealed by a SQUID-like SQI at high-displacement field) does not produce an even-odd modulation. The probability of these two observations being a coincidence for one particular sample is already very low. Regardless, in supplementary sections 5 and 7, we added a second set of measurements in a second device that exhibits exactly the same phenomenology between even-odd SQI and band-inversion. Additional control measurements in a non-encapsulated device lend further support to our interpretation (See Fig. 5d-f in the manuscript).

We have taken great care to present the above reasoning more clearly, factually and precisely in the revised version. We hope the reviewer will find it satisfactory.

I don't understand the sentence "A higher D, I_c decreases monotonically due to a gradual

opening of a bandgap". Firstly, there is no bandgap signature in the data. The resistance is 2 kOhms, and I see no evidence of activation (Arrhenius plot?).

We thank the referee for this comment. We were not precise enough, and have now improved our discussion. Indeed, in the absence of edge modes of any kind, a bandgap would imply normal resistances much above the von Klitzing constant. The measurement merely show that resistance in the BIP monotonically grows with $|D|$, and that the $B_z = 0$ critical current decreases. To be precise, this is not enough to claim an "insulating behavior". The magnitude of the resistance is too low. We believe the reason, as discussed above, is that a significant residual bulk transport mediated by non-percolating charge puddles exists in our samples. The Arrhenius plots (see Fig. S3 in the supplementary information) show that activated transport is cut-off low temperatures by a resistance ceiling due to this bulk conduction (and probably also to the presence of trivial vacuum edge states for all $|D|$). However, this ceiling quickly grows as $|D|$ is increased, as expected from a gradual depletion of the puddles caused by an increasing BLG gap around neutrality. Eventually, at $D > 150\text{meV/nm}$, the SQI becomes SQUID-like, suggesting that all bulk puddles are essentially depleted, and only uncoupled-edge-transport remains through trivial vacuum modes.

In addition to the new Arrhenius plots now included in the supplementary material (section 2B), we have adapted our description of the BIP in the main text to clarify the role of puddles and the limitations to activated transport.

Secondly, aren't we simply observing a constant $I_c R_n$, irrespective of the physics involved in varying the resistance? The sentence "The presence of a supercurrent and its suppression at $D=0$ is a signature of Josephson coupling across a BLG in an inverted phase". I'm sorry, that's not correct. This observation is just the signature of a constant $I_c R_n$. It in no way indicates "Josephson coupling through a phase-reversed BLG". The causality is wrong.

This is a valid point: in itself, the increase of critical current (and the decreased resistance) does not prove that bands have been inverted. If we understand "band inversion" as a topological band reconnection, demonstrating a band inversion would require showing that the gap has become topological. So the causality implied by our wording was indeed wrong.

To be precise, our evidence for an inverted phase is encoded in the even-odd modulation of the IGP, as already discussed, and hence we have to defer the claim of a band inversion when presenting this data. We have fixed this in the updated text, which now reads “The suppression of the supercurrent around $D = 0$ is a consequence of an phase with higher resistance”

Overall the BLG system is definitely interesting and the Fraunhofer data are beautiful. I strongly encourage the authors to modify their writing with more humble claims regarding the signature/presence of helical edge states, which I think weakens the paper.

We have carefully revised our discussion with this and other related comment in mind (see manuscript text in red), and trust the reviewer will find the new version acceptable.

Few more comments: - In the paragraph “Before proceeding to the superconducting [...]” I suggest to add the reference Fig.2a, which appears only at the end, earlier in the paragraph.

We have added the reference in second sentence of the paragraph: “As a first step we measure the dependence of normal resistance with bottom and top gates, which independently control the carrier density, n , and the displacement field, D (Fig. 2a).”.

Figure 5: I suggest to plot the amplitude of each lobes as a function of D for both cases.

We have extracted the amplitude of few lobes as a function of D for Dev B and Dev C from Fig. 5 presented in the manuscript and plotted in Fig. 4 further below in this reply. The amplitudes of successive lobes (starting from 0 to 3) for Dev C gradually decrease at all D . Similar trend is also observed for Dev B at higher values of D . Inside IGP (close to $D = 0$), the amplitude of even lobe (0, 2, 4) is always higher than the next odd lobe (1, 3, 5), implying the presence of even-odd effect. The even-odd effect, however, is more difficult to appreciate when plotting the data in this fashion.

Concerning the effective area and the model, I understand even-odd effect implies a $2\Phi_0 = h/e$ periodicity of the Fraunhofer pattern due to a cross Andreev conversion along

FIG. 4. The critical current amplitude for different lobes as a function of D for Dev B (a) and Dev C (b). The values represent the averaged amplitude for a lobe on positive and negative field directions. The vertical dotted line in (a) represent the inversion point.

the superconducting contact (Fig S8). The theoretical part explains that this circular trajectory picks up an Aharonov-Bohm phase (so the $h/2e$ periodicity). Then I would expect that this component of quantum interference has the effective area equal to the BLG area and not the larger effective area due to the magnetic penetration depth. Can the authors comment on that?

Yes, definitely. The effective area as extracted from the Fraunhofer is larger than the BLG area, but this correction is due, rather, to a significant field focusing into the BLG region as observed in various JJs [Physica C 367, 229 (2002), Nano Lett. 15, 1803 (2015), Phys. Rev. B 95, 035307 (2017)]. The Meissner effect in the superconductors expels the fields, which then become deflected towards the normal BLG region. The magnetic flux increases, so that the Fraunhofer pattern varies more rapidly with magnetic field, and the effective area increases.

We have added a discussion about flux-focusing in section 3C of the supplementary material.

REVIEWER 3

Rout et al. investigate a Josephson junction (JJs) of bilayer graphene encapsulated in WSe2 with top and bottom gates. In the normal state bilayer graphene in contact with WSe2 has previously been shown to host a helical edge state due to Ising spin-orbit. Here, the authors first investigate the normal state properties of a large device showing that, close to charge neutrality, a finite displacement field results in an initial decrease followed by an increase in resistance of the device, which is characteristic of the bulk band inversion. They also show that an in-plane field modifies the magnetoresistance only for displacement field $D \approx 0$. Moving to JJs, Rout et al. show that the critical current at charge neutrality exhibits a two peak structure as a function of displacement field, again characteristic of the bulk band inversion. Finally, the main result of the paper is that the Fraunhofer pattern of the JJs show an even-odd pattern only for $D \approx 0$, which the authors ascribe to the presence of a helical edge. This is backed up by numerical simulations and by a control experiment on a device that is not encapsulated in WSe2 and which does not show this even-odd pattern.

Overall I find that the results are very well written, logically presented, and interesting. The topic of inducing superconductivity in helical or chiral edges is of significant recent interest and the paper invites many avenues for investigating the nature of the induced superconductivity in this setup.

I am minded towards eventual publication in Nat. Comm., but I reserve final judgment until the authors have responded to the following points:

- 1) My main concern is that the key result, the odd-even pattern of the Fraunhofer in a ballistic JJ with IGP, is only shown for a single device (dev B). I appreciate that fabrication can be time consuming, but I would be a lot more comfortable if the results were reproduced on (at least) a second device. As it currently stands I do not think I can recommend acceptance of a paper based on results from only a single device, even if the current results are very convincing.

We thank the reviewer for this comment. We had taken a limited data set on the band inversion and even-odd effect in one other junction (Dev D). This was omitted in the previous version, but is now included. The results are in good agreement with the previous measurements. We present the gate map and SQI measurements at different D values for

Dev D in the supplementary file.

2) In Fig. 2c the authors show the magnetoresistance is negative only close to zero in-plane magnetic field and zero displacement field, arguing that this is due to the opening of a gap in the helical dispersion due to the field. However, the effect is very small ($\sim 2\%$).a) Can the authors comment on the small size of this effect? Naively one would expect a large increase in resistance due to the opening of a gap unless there are many other conduction channels than just the helical edge.

The reason is that there is a sizeable background of bulk conduction, that is added to the current flowing along the edge modes: the sheet resistance ($2.7 \text{ k}\Omega$ at $D = 0$) for Dev A is much smaller than the expected value of $h/4e^2$ from four edge states (See Fig. 1). This extra bulk current reduces the relative change in resistance with B_y . Its existence and possible origin is now discussed in detail in the revised manuscript. Unlike the magnetoresistance, the SQI measurements are sensitive to the spatial profile of currents, and are thus much more sensitive to the effect of helical edge modes. Nevertheless, the behavior of normal magnetoresistance with in-plane fields offers valuable information, since it addresses the spin structure of the edge modes. The negative MR lends support to the interpretation that there are indeed helical modes that come with the inverted gap.

b) I appreciate that Fig. 2 is a larger device, but if there are several other conduction channels, why is the size of the Fraunhofer even-odd effect in dev B so large (almost killing the second lobe at $D=0$)? Does this not contradict somewhat the theoretical analysis in Fig. 4d-e that shows only a modest even-odd effect when several channels are present.

This is an astute observation. The magnitude of the even-odd effect for the case of N high-transmission trivial channels produced by a single low-transmission helical channel should be small. However, for $N = 0$, an increasing transmission of the helical mode also increases the even-odd effect (see Fig. 4b in the manuscript), which can become arbitrarily large close to perfect transmission. In the presence of $N > 0$ trivial high-transmission modes the effect is not so strong, but can still be significant. Finally, if N is around 2 to 6 but these trivial modes have moderate or low transmission, and the two helical modes per edge

have high transmission, it is quite easy to have the kind of even-odd modulations seen in the experiment

This is in fact another argument consistent with a helical nature of the extra edge modes in the IGP. Helical edge modes typically have a higher transmission than trivial edge modes, since helical-mode backscattering on non-magnetic disorder is suppressed by spin-conservation. Hence, their even-odd contribution is enhanced. Despite the potential interest of this argument, we have decided not to include it in the manuscript since we have no alternative method to assess the transmission distribution of different edge modes.

c) It would be useful (and convincing) to include in the supplemental material a similar plot to Fig. 2, but at a very small but finite density, such that the doping of the helical edge is finite and the chemical potential larger than any induced magnetic gap. In such a scenario one would expect very little change in MR.

We agree with the reviewer that the observation of nearly zero MR at a very small n would further strengthen our claim of helical edge states. Regretably, however, we do not have this measurement and are unable to present it.

3) In Fig. 3 it is shown that the displacement field results in two peaks in the critical current, I_c , which correspond to the expected transition from IGP. What is the impact of an in-plane magnetic field on I_c ? In particular within the IGP.

We have checked the effect of in-plane field for our JJs. In Fig. 5 attached below, we present $I_c(D)$ at an in-plane field of 120 mT. The Zeeman field reduces the critical current for all D values. However, no distinguishable difference in the suppression factor can be observed across the band inversion. Beyond 120 mT, the critical current measurements (as well SQI) were not stable due to random vortex jumps and flux pinning inside the superconducting contacts. Therefore we can not measure at higher fields, where no helical edges are expected.

4) On a similar note, since the application of an in-plane field gaps out the helical modes it should therefore remove the even-odd pattern in the Fraunhofer lobes. Is this the case? Such a restoration of the usual Fraunhofer pattern would be a very convincing demonstra-

FIG. 5. Current-bias dependence of the differential resistance dV/dI for Dev B as a function of D at $n = 0$ and in-plane magnetic field $B_y = 120$ mT. The black line shows the critical current for $B_y = 0$ mT [also presented in Fig. 5(a)].

tion of the nature of the the even-odd pattern stemming from the helical edge.

That is an excellent idea. We actually tried this. The presence of helical edge states could potentially be confirmed by analysing the impact of an in-plane magnetic field in the SQI. Unfortunately, however, we were unable to go beyond 120mT in the superconducting phase without encountering device instabilities, as explained above. The maximum stable fields were found to be insufficient to suppress the even-odd modulation visibly, which led us to suspect that the effect of B_y on our devices in the superconducting phase is considerably more involved than in the normal phase.

5) In Fig. S7 comparing the Fraunhofer at large and zero displacement field it is clear that the BLG (Fig. S7f) exhibits a shift of the lobes as function of displacement field (which is also visible in Fig. S7e). However, this shift does not appear for the WSe2 encapsulated sample. Can the authors comment on what might have caused the shift for the BLG sample, but why it is not present for the sample with an IGP?

This may be a visual artifact. Our understanding is as follows. The apparent squeeze-

ing of the SQI period is due to the transition of the Fraunhofer $I_c \sim \text{sinc}(\pi\Phi/\Phi_0) = \sin(\pi\Phi/\Phi_0)/(\pi\Phi/\Phi_0)$ profile around $D = 0$ (with finite bulk conduction) to the SQUID profile $I_c \sim \cos(\pi\Phi/\Phi_0)$ at large D (with only trivial-edge-mode conduction). The central peak of the sinc function has a doubled width $2\Phi_0$, while all lobes in the cos function have width Φ_0 . The Fraunhofer-to-SQUID transition as a function of D happens in a similar way in Dev B and Dev C, however. The range of displacement fields D plotted in Fig. S7 reaches higher values for Dev C than for Dev B, which might give the impression that the "squeezing" is more pronounced without WSe2 encapsulation.

Minor comment: In Fig. 4b and e the colour map is dominated by the central lobe of the Fraunhofer pattern, which makes it a bit difficult to observe the main feature of interest (the side lobes). I would recommend setting adjusting this map to emphasise the side lobes.

We thank the reviewer for pointing out this issue with color map. We have adjusted it to show the even-odd effect more clearly.

LIST OF CHANGES

The different changes and additions mentioned in each reply are marked in red, both in the main text and the supplementary information. We have adjusted the discussion to be more precise and factual. As part of this change, we have introduced an explicit discussion about the background bulk conduction, its probable origin and its impact on normal resistance (including its temperature dependence) and the SQI patterns. We have also extended the Supplementary material with additional experimental data, analysis and theory. In addition, a correction was made to Fig 3 which had an incorrect scale.

REVIEWERS' COMMENTS

Reviewer #2 (Remarks to the Author):

The authors have significantly toned down their claim about the signature of helical edge transport in the resistance. As I stated in my first report, I repeat that their work is clearly relevant for the field of topological superconductivity and their data are of high quality. Their article is now more factual with a fair discussion on the issue of the low and non-quantized resistance in the inverted gap phase. They also answered all my other concerns and questions. I therefore highly recommend the publication of this work in Nature Communications.

Minor comment: when discussing activation behavior in the SI, I don't think it is correct to mention variable range hopping, nor activated transport if $R < h/e^2$ for a 2D system. I would just mention parallel bulk conduction channels that somehow connects (by percolation ?) the two electrodes. I understand it is frustrating but the bulk is somehow conducting!

Reviewer #3 (Remarks to the Author):

Rout et al. have provided convincing answers/data that satisfy most of my questions. I also understand that such devices are highly complex and very much appreciate their honesty when data is not available/does not fit a naive theory perspective.

I recommend publication, although I agree with both of the other reviewers that some of the language around the main claims should be tempered. For instance in the abstract “confirms this interpretation” would be better replaced by “supports” etc.

Finally, I think it would be useful to add a comment on the links to recent papers showing negative downstream resistance due to crossed Andreev reflection resulting from induced superconductivity in topological edge states e.g. G.-H. Lee, et al., Nat. Phys. 13, 693 (2017) (more recent papers are also discussed in https://www.condmatclub.org/jccm_july_2023_01/)

We thank both referees for their positive reports. A point by point response to all their comments is provided below. All changes are marked in red in the main text, including those required directly by the editor.

REVIEWER 2

The authors have significantly toned down their claim about the signature of helical edge transport in the resistance. As I stated in my first report, I repeat that their work is clearly relevant for the field of topological superconductivity and their data are of high quality. Their article is now more factual with a fair discussion on the issue of the low and non-quantized resistance in the inverted gap phase. They also answered all my other concerns and questions. I therefore highly recommend the publication of this work in Nature Communications.

We thank the referee for the positive recommendation.

Minor comment: when discussing activation behavior in the SI, I don't think it is correct to mention variable range hopping, nor activated transport if $R < h/e^2$ for a 2D system. I would just mention parallel bulk conduction channels that somehow connects (by percolation ?) the two electrodes. I understand it is frustrating but the bulk is somehow conducting!

We agree with the reviewer: the analysis of thermal activation behavior using simplistic hopping or Arrhenius models is probably not rigorous enough due to the presence of a background bulk conduction. Therefore we have decided to remove the discussion about hopping mechanisms in the SI section 2B. To clarify this point we have also modified a sentence in the main text (marked in red on page 2) and removed the sentence "Hopping between the puddles lowers the resistance at $n = 0$, while the presence of helical edge modes (to be discussed shortly) flowing around our samples in the IGP, suggests that these puddles are non-percolating with moderate hopping."

REVIEWER 3

Rout et al. have provided convincing answers/data that satisfy most of my questions. I also understand that such devices are highly complex and very much appreciate their

honesty when data is not available/does not fit a naive theory perspective.

I recommend publication, although I agree with both of the other reviewers that some of the language around the main claims should be tempered. For instance in the abstract “confirms this interpretation” would be better replaced by “supports” etc.

We thank the referee for the positive comments and recommendation. Regarding further tweaks to the language, we have replaced ”confirms” with ”supports” (marked in red) as suggested. We have also removed the last sentence of the abstract, which was unnecessary.

Finally, I think it would be useful to add a comment on the links to recent papers showing negative downstream resistance due to crossed Andreev reflection resulting from induced superconductivity in topological edge states e.g. G.-H. Lee, et al., Nat. Phys. 13, 693 (2017) (more recent papers are also discussed in https://www.condmatjclub.org/jccm_july_2023_01/)

We appreciate this comment. However, given that we present no results on downstream resistance measurements in our experiment, and that explaining the connection would require some extra space and a diversion from the central narrative, we would prefer to not go into this discussion in this work.